# Loop-Mediated Isothermal Amplification of *Trypanosoma cruzi* DNA for Point-of-Care Follow-Up of Anti-Parasitic Treatment of Chagas Disease

**DOI:** 10.3390/microorganisms10050909

**Published:** 2022-04-26

**Authors:** Arturo A. Muñoz-Calderón, Susana A. Besuschio, Season Wong, Marisa Fernández, Lady J. García Cáceres, Patricia Giorgio, Laura A. Barcan, Cole Markham, Yanwen E. Liu, Belkisyole Alarcón de Noya, Silvia A. Longhi, Alejandro G. Schijman

**Affiliations:** 1Laboratorio de Biología Molecular de la Enfermedad de Chagas, Instituto de Investigaciones en Ingeniería Genética y Biología Molecular (INGEBI-CONICET), Buenos Aires 1428, Argentina; arturomc35@gmail.com (A.A.M.-C.); bsusanaalicia@gmail.com (S.A.B.); ladyjuliette1108@gmail.com (L.J.G.C.); longhi.ingebi@gmail.com (S.A.L.); 2AI Biosciences, Inc., College Station, TX 77845, USA; season.wong@aibiosciences.com (S.W.); markhamc@gmail.com (C.M.); yanwenliu1997@gmail.com (Y.E.L.); 3Hospital de Enfermedades Infecciosas “Dr. Francisco J. Muñiz”, Buenos Aires 1282, Argentina; marisa.fernandez@gmail.com; 4Servicio de Infectología, Hospital Británico de Buenos Aires, Buenos Aires 1280, Argentina; patrilougiorgio@gmail.com; 5Sección Infectología, Departamento de Medicina, Hospital Italiano, Buenos Aires 1199, Argentina; labarcan@gmail.com; 6Instituto de Medicina Tropical, Universidad Central de Venezuela, Caracas 1053, Venezuela; belkisuole@gmail.com

**Keywords:** loop-mediated isothermal amplification, real-time PCR, *Trypanosoma cruzi*, Chagas-HIV, orally transmitted Chagas disease, primary infection after transplant in seropositive donor-seronegative recipients, Chagas disease reactivation

## Abstract

A loop-mediated isothermal amplification assay was evaluated as a surrogate marker of treatment failure in Chagas disease (CD). A convenience series of 18 acute or reactivated CD patients who received anti-parasitic treatment with benznidazole was selected—namely, nine orally infected patients: three people living with HIV and CD reactivation, five chronic CD recipients with reactivation after organ transplantation and one seronegative recipient of a kidney and liver transplant from a CD donor. Fifty-four archival samples (venous blood treated with EDTA or guanidinium hydrochloride-EDTA buffer and cerebrospinal fluid) were extracted using a Spin-column manual kit and tested by *T. cruzi* Loopamp kit (Tc-LAMP, index test) and standardized real-time PCR (qPCR, comparator test). Of them, 23 samples were also extracted using a novel repurposed 3D printer designed for point-of-care DNA extraction (PrintrLab). The agreement between methods was estimated by Cohen’s kappa index and Bland–Altman plot analysis. The *T. cruzi* Loopamp kit was as sensitive as qPCR for detecting parasite DNA in samples with parasite loads higher than 0.5 parasite equivalents/mL and infected with different discrete typing units. The agreement between qPCR and Tc-LAMP (Spin-column) or Tc-LAMP (PrintrLab) was excellent, with a mean difference of 0.02 [CI = −0.58–0.62] and −0.04 [CI = −0.45–0.37] and a Cohen’s kappa coefficient of 0.78 [CI = 0.60–0.96] and 0.90 [CI = 0.71 to 1.00], respectively. These findings encourage prospective field studies to validate the use of LAMP as a surrogate marker of treatment failure in CD.

## 1. Introduction

Chagas disease (CD), a neglected tropical disease (NTD) caused by the protozoan *Trypanosoma cruzi*, affects about 7 million people worldwide, mainly in endemic areas of Latin America [1]. Transmission may occur by different routes, such as vector-borne, oral, congenital, transfusional or organ transplant and laboratory accidents. The disease evolves from an acute to a chronic phase that may develop in up to 30% of cases to cardiac disease and in 10% to digestive mega-syndromes, neurological and/or mixed complications [2,3]. A proportion of immunocompromised chronic CD patients associated with HIV, organ transplantation, autoimmune disease or oncologic treatments may experience disease reactivation, which usually develops in severe clinical forms with high parasitaemia [4,5]. Thus, early diagnosis of acute and reactivation infections is relevant because it allows early trypanocidal treatment, avoiding severe clinical presentations. Indeed, monitoring of parasitological response to treatment by means of surrogate markers may guarantee more accurate follow-up and earlier detection of treatment failure. However, we still lack sensitive surrogate markers of treatment failure that can be applied with simple laboratory manipulations and inexpensive equipment [6,7]. Loop-mediated isothermal amplification (LAMP) has a potential to become a tool for monitoring anti-parasitic treatment. To date, no studies have tested LAMP for CD treatment follow-up [8,9]. In this context, we used LAMP to analyse a series of archival clinical samples collected from CD patients who received benznidazole to assess its ability to detect *T. cruzi* DNA, which indicates treatment failure, and compared it with standardized qPCR, which is currently used in clinical trials and in clinical practice [6]. Moreover, in a subset of samples, we used a novel repurposed 3D printer for a rapid automatic DNA extraction and purification for downstream LAMP testing in low-resource settings.

## 2. Materials and Methods

Clinical groups.

(i)Orally transmitted Chagas Disease (oCD): blood samples collected from nine patients residing in Chichiriviche de la Costa, Vargas State, Venezuela, diagnosed with oCD after consumption of *T. cruzi*-contaminated guava juice in March 2009 [10]. Parasitological diagnosis was made either by microscopic search of trypomastigote forms in peripheral blood or by parasite culture, and serological analysis by in-house assays (ELISA and IHA) with a *T. cruzi* epimastigotes delipidized antigen for detection of anti-human IgG and IgM [10]. Clinical examination included electrocardiogram (EKG) and echocardiography (ECHO). For molecular diagnosis, 5 mL blood samples were collected in 5 mL of guanidine hydrochloride 6 M-EDTA 0.2 M, pH 8.00 (GE), and stored at 4 °C.(ii)HIV-Reactivated Chagas Disease (HIV-RCD): EDTA-treated blood and/or cerebrospinal fluid (CSF) samples were recovered from three *T. cruzi* patients with HIV coinfection diagnosed by central nervous system (CNS) CD reactivation. The patients were admitted and clinically monitored between 2014 and 2018. Diagnosis included microscopic analysis of CSF specimens, Strout test and central nervous system imaging. The samples were collected and stored at −20 °C.(iii)Chronic Chagas disease transplanted recipients with reactivation (Tx-RCD). EDTA blood samples from five chronic CD patients who underwent organ transplantation and presented reactivation of *T. cruzi* infection due to immunosuppressive treatments. All patients received standard etiological treatment (benznidazole 5 mg/kg/day for 60 days), except for Tx-RCD patient 4, who received a half-dose regime for 7 days to avoid renal failure and, due to the persistence of the parasite load, continued with the conventional treatment mentioned above.(iv)Recipient of organs from an infected donor (Tx-RID). A seronegative recipient of a kidney and liver transplant from a *T. cruzi*-infected donor who became infected after transplant in 2016. This primary infection was diagnosed by qPCR as described elsewhere [11]. The patient received supervised treatment with 5–7 mg/kg/day of benznidazole for 60 days. Five millilitres of blood were collected in EDTA tubes and stored at −20 °C until processing for qPCR and Tc-LAMP.

DNA extraction from clinical specimens for qPCR and LAMP: Manual DNA extraction was carried out from 200 µL EDTA blood or CSF samples, or 300 µL of GE blood using the High Pure PCR Template Preparation Kit (Roche Diagnostics GmbH, Mannheim, Germany) as recommended by the manufacturer [9]. In a set of 23 samples (19 EDTA blood and 4 GE blood), DNA was also purified using the MagMax Multi Sample Ultra 12.0 DNA extraction kit (Thermo Fisher Scientific Inc, Waltham, MA, USA) in a repurposed 3D printer (PrintrLab extraction device version 4.0, AI Biosciences, Inc., College Station, TX, USA) designed for rapid and low-cost point-of-care molecular diagnostics (Figure 1) [12]. The procedure was optimized for 200 μL of starting blood. Briefly, samples, lysis buffer and magnetic particles were loaded in the first row. Subsequently, the DNA bound to the magnetic particles was captured and transferred to successive rows for the washing steps and the final DNA elution step (Figure 1).

Every DNA-extraction experiment included a negative DNA-extraction control, which was a sample of peripheral blood from a *T. cruzi*-seronegative individual. The DNA extracts were stored at −20 °C until their use for qPCR or LAMP.

*Trypanosoma cruzi* Loopamp kit: Tc-LAMP (Eiken Chemical Co., Ltd., Tokyo, Japan) targets the repetitive satellite DNA sequence of *T. cruzi* [8]. The LAMP reaction used 30 μL of DNA eluates that were incubated at 65 °C for 40 min, followed by a step at 80 °C for five minutes for enzyme inactivation, as reported in [9]. Results were observed by the naked eye or by UV visualization using a P51^TM^ molecular fluorescence viewer with yellow filter [8,9] and expressed as positive or negative (Figure 2). Two LAMP replicates were carried out per DNA extract. Sample panels were masked, and a series of six samples each were randomly chosen to perform a LAMP round that included a non-template control provided by the manufacturer and a positive control (PC: 30 µL of 1fg/μL CL-Brener stock DNA, not provided). The LAMP operator read amplification results blinded to qPCR findings and clinical data.

Multiplex real-time quantitative PCR (qPCR): Duplex qPCR using TaqMan probes targeted to *T. cruzi* satellite DNA plus an internal amplification control (IAC) was carried out in duplicates in a Rotor Q thermocycler (Qiagen) following standardized conditions [7,11]. In the series of oCD and Tx-RCD samples, human RNAse P gene-based amplification was used as internal control of DNA integrity. All the samples were checked for qPCR inhibitors using the criterion of Tukey, and quantification of parasite loads was performed using standard curves as reported in [11]. Laboratory operators of the qPCR assay were blinded to index test results. The research evaluator had no access to Tc-LAMP results or to clinical information before the end of sample processing and reporting.

Statistical analysis: *T. cruzi* Loopamp kit (Tc-LAMP) was compared with standardized qPCR using panels of archival clinical specimens collected from the above-mentioned groups of patients.

The Cohen’s kappa coefficient (K) and the Bland–Altman plot were used to measure the agreement between Tc-LAMP (Spin-column and PrintrLab DNA extractions) and qPCR results obtained in paired samples. All the analyses were performed with the RStudio Team software [13]. A *p*-value of <0.05 was considered statistically significant.

## 3. Results

Tc-LAMP results were observed by the naked eye or by UV visualization, as shown in Figure 2. Both visualization methods were considered for reporting the results. There were no disagreements between them in the samples tested. Moreover, there was no disagreement between duplicate amplifications from a same DNA extract.

oCD: 24 GEB samples from nine patients with oCD (eight paediatric patients and one adult) were included. All of them were infected with Tc I parasite populations, and qPCR monitoring at 12, 36, 60 or 108 months after treatment indicated treatment failure in most patients (oCD group (Table 1). These 24 samples were tested by Tc-LAMP (Spin-column), and 4 of them also underwent Tc-LAMP (PrintrLab) testing. The results of the Tc-LAMP (PrintrLab) were concordant with those of qPCR. For Tc-LAMP (Spin-column) testing, the results were concordant with qPCR in 22 cases, and 2 cases were qPCR positive and LAMP negative. One of the Tc-LAMP (Spin-column) negative samples was positive by Tc-LAMP (PrintrLab) (oCD6, Table 1).

HIV-RCD: A total of nine samples were tested by Tc-LAMP (Spin-column) and qPCR in peripheral blood and/or CSF samples and only two samples were available for PrintrLab DNA extraction. At the time CD reactivation was diagnosed, the corresponding samples (EDTA blood and CSF) tested by Tc-LAMP (Spin-column) and qPCR had concordant results (HIV-RCD group, Table 1). Eight post-treatment samples were tested, and treatment response detected by qPCR and Tc-LAMP (Spin-column) was similar (Table 1). A discordant finding (Tc-LAMP-positive, qPCR-negative) was detected in the blood sample from case HIV-RCD 3, collected 38 days after treatment had started. Interestingly, a CSF sample collected 7 days after initiation of therapy had a high parasite load. Whereas 1 day after, no parasite loads were detected in the blood and remained undetected in the last sample available, which was collected 14 days after treatment initiation and 2 days before the patient died.

Tx-RCD: Five Tx-RCD patients were included. Patients Tx-RCD 1 and Tx-RCD 4 exhibited a two-log increase in parasite burden at days 25 and 14 after orthotopic kidney transplantation, respectively. In the first case, the patient received standard etiological treatment (benznidazole 5 mg/kg/day for 60 days), and parasite load reduction was observed seven days after treatment was initiated. In the second case, treatment with benznidazole was initiated based on a positive Strout test, which is the gold standard parasitological method for the diagnosis of CDR. However, qPCR was positive thirteen days earlier than the Strout test. To avoid renal failure, this patient received half the standard dose for 7 days. However, this dose was not sufficient since qPCR showed an increase in parasitaemia, and thus, the patient was treated with a higher dose, showing a therapeutic response with a non-detectable qPCR result 21 days after the high dose treatment. Patient Tx-RCD 2 underwent orthotopic heart transplantation to treat end-stage chronic Chagas heart disease. Nine months after transplantation, a parasite load of 24.41 par.eq/mL was estimated by qPCR. The patient received anti-trypanosomal chemotherapy and after completing 60 days of treatment, the parasite load measured by qPCR was below the limit of detection, suggesting therapeutical response. Patient Tx-RCD 3 received orthotopic liver transplant and was monitored by Strout test and qPCR. The qPCR-based follow-up after transplantation showed parasite loads between 1.09 and 12.11 par.eq/mL, whereas Strout test detected patent parasitaemia 78 days after transplant (Tx). The latter finding supported the initiation of anti-parasitic treatment. Two weeks after treatment started, both Strout test and qPCR became non-detectable, suggesting a favourable parasitological response. Patient Tx-RCD 5 presented multiple myeloma and CD reactivation due to chemotherapy before undergoing autologous stem cell transplantation. The sample withdrawn two days before Tx gave a mean parasite load of 54.25 par.eq/mL. After Tx, the patient received anti-parasitic treatment, and parasitaemia decreased to 2.41 par.eq/mL one week later, reaching non-detectable levels in the sample tested 22 days after the autograft.

In these five patients, the presence of *T. cruzi* DNA was also tested by Tc-LAMP (Spin-column) and LAMP (PrintrLab) in 12 EDTA blood samples. Five samples were tested after Tx and seven samples once treatment was initiated (Table 2). The results of pre-treatment samples were all positive for qPCR and Tc-LAMP (both DNA extraction methods), whereas, five of seven post-treatment samples had concordant results, one was only positive by Tc-LAMP (PrintrLab) (Tx-RCD 1.5) and another one was only positive by Tc-LAMP (Spin-column) (Tx-RCD 3.5).

Tx-RID: This case corresponds to a non-infected patient who received liver and kidney transplantation and was followed-up by qPCR. After detecting an increase in parasite load from 0.49 to 39.68 par.eq/mL, the patient was diagnosed with primary *T. cruzi* infection. Consequently, he received anti-parasitic therapy for three weeks. Six EDTA blood samples (two pre-treatment and four post-treatment) were tested by Tc-LAMP (Spin-column). Two samples (one pre-treatment and one post-treatment) were also tested by Tc-LAMP (PrintrLab) (Table 2). Tc-LAMP (Spin-column) was negative in the sample with 0.49 par.eq/mL and became positive in the sample obtained 61 days after Tx, in agreement with Tc-LAMP (PrintrLab). After treatment was initiated, the patient became qPCR negative in the sample tested eight days later and persisted negative during follow-up for 183 days after treatment, suggesting a favourable response, in agreement with Tc-LAMP results.

Overall agreement between qPCR and Tc-LAMP (Spin-column, *n* = 54) or Tc-LAMP (PrintrLab, *n* = 23) in paired samples showed a Cohen’s kappa coefficient of 0.78 [CI = 0.60–0.96] and 0.90 [CI = 0.71 to 1.00], respectively. The agreement between Tc-LAMP (Spin-column) and Tc-LAMP (PrintrLab) in 23 paired samples was 0.704 [0.39 to 1.00]. Additionally, the mean difference for both Tc-LAMP data groups compared with the comparator qPCR assay was 0.02 [CI = −0.58–0.62] and −0.04 [CI = −0.45–0.37], respectively, demonstrating the high concordance of the diagnostic methods evaluated (Bland–Altman analysis, Figure 3).

## 4. Discussion

Very few studies of NTDs have evaluated LAMP for monitoring the effectiveness of chemotherapy, such as *Schistosoma japonicum* infection [14]. To our knowledge, this is the first report that evaluates LAMP for assessment of treatment efficacy in CD patients [15]. Our interest in evaluating LAMP for monitoring CD treatment is based on the fact that this NTD is highly endemic and that most patients live in resource-limited settings with poor investment in public health policies. Noteworthily, the *T. cruzi* Loopamp assay showed excellent agreement with the comparator qPCR test. This LAMP assay could detect samples infected with different discrete typing units, such as Tc I in oCD patients from Venezuela [16] and Tc V and Tc VI in immunosuppressed patients [17,18]. Moreover, it performed well in DNA samples obtained in different supports and stabilizing agents, such as frozen blood treated with EDTA or GE buffer, as well as in CSF specimens. Tc-LAMP detected samples with more than 0.5 par.eq/mL, which is the approximate limit of detection of the comparator standardized qPCR test (0.69 par.eq/mL for Tc VI CL-Brener clone) [11]. Discordant Tc-LAMP-negative/qPCR-positive findings were detected in samples with loads around that limit of detection, while there were three post-treatment samples with non-detectable amplification by qPCR and positive Tc-LAMP results (HIV-RCD 3, Table 1 and Tx-RCD 1 and 3, Table 2), suggesting that Tc-LAMP could be a very sensitive indicator of treatment failure. This high sensitivity makes it potentially useful for monitoring chronic CD patients receiving treatment and for early diagnosis of congenital Chagas disease in infants [19,20,21].

A novel procedure was optimized, using the 3D PrintrLab extractor device [12] (Figure 1) coupled to Tc-LAMP, with high agreement with Tc-LAMP starting from Spin-column extracted DNA and qPCR. Its advantage includes the possibility of rapid results: DNA extraction may take about 40 min, and amplification takes another 40 min, followed by immediate naked-eye or UV-light-mediated visualization (Figure 2). The approximate cost of the PrintrLab device is approximately ten times lower than that of most automatic DNA extraction robots, while the cost of Tc-LAMP is at least 50% lower than that of qPCR.

In summary, this report promotes future prospective studies for validating Tc-LAMP in the field for early assessment of treatment response in CD using point-of-care DNA extraction and amplification techniques that may expand the implementation of molecular diagnostics and monitoring of CD to low-resource settings in endemic areas.

## Figures and Tables

**Figure 1 microorganisms-10-00909-f001:**
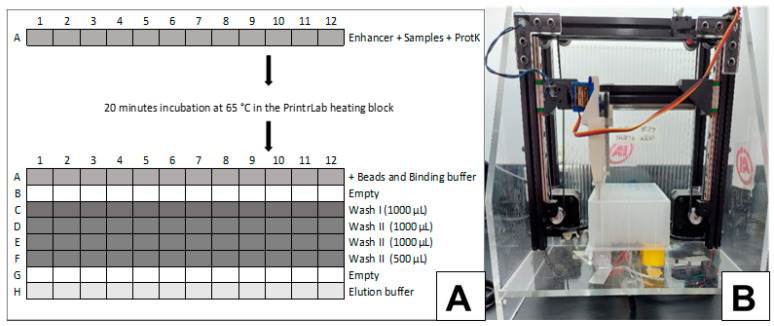
A plate-layout scheme for the PrintrLab DNA-extraction procedure. Blood samples, lysis buffer and magnetic particles are loaded in row (**A**). Subsequently, the DNA bound to the magnetic particles is captured and transferred to successive rows for the washing steps and the final DNA elution step. All steps are performed with agitation and slow capture of the beads. (**B**) PrintrLab extraction device.

**Figure 2 microorganisms-10-00909-f002:**
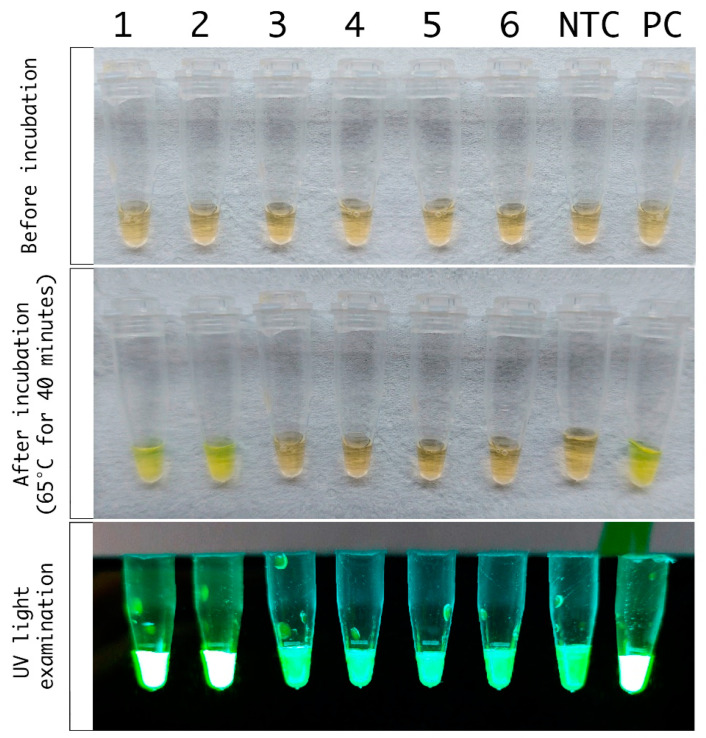
Tc-LAMP PrintrLab DNA extraction of samples from case Tx-RCD 5. Panel A: Strip of Tc-LAMP microtubes before amplification. Panel B: Naked-eye visualization of Tc-LAMP microtubes after amplification. Panel C: Visualization using the P51^TM^ molecular fluorescence viewer with yellow filter. Wells 1 and 2: duplicate Tc-LAMP from sample collected the day treatment was initiated (Tx-RCD 5.1). 3 and 4: duplicate Tc-LAMP from sample collected 8 days after treatment initiation (Tx-RCD 5.2). 5 and 6: duplicate Tc-LAMP from sample collected 24 days after treatment initiation (Tx-RCD 5.3). NTC: Non-Template Tc-LAMP control, PC: Positive *T. cruzi* DNA control.

**Figure 3 microorganisms-10-00909-f003:**
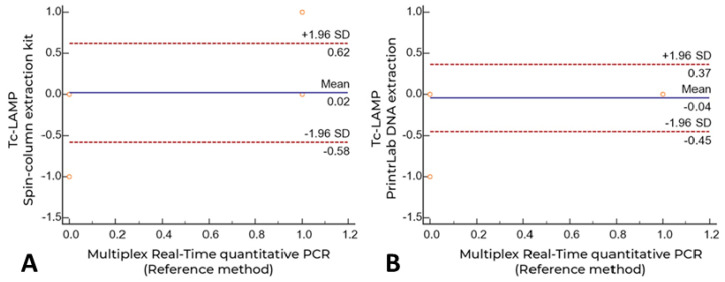
Agreement between multiplex real-time quantitative PCR and Tc-LAMP (Spin-column and PrintrLab DNA extractions) by Bland Altman plot analysis. (**A**) Manual extraction using the High Pure PCR Template Preparation Kit (Spin-column); (**B**) PrintrLab DNA extraction device version 4.0 using the MagMax DNA extraction kit.

**Table 1 microorganisms-10-00909-t001:** Follow-up of oCD and HIV-RCD clinical groups by qPCR and Tc-LAMP.

Clinical Group	ID Patient	Gender (F/M)	Age (years)	Pathology/Laboratory/CNS Imaging Findings at Diagnosis	Sample Type	Sample ID	Time from Treatment Initiation	^#^ Mean Parasite Load (par.eq/mL)	^§^ qPCR Result	Tc-LAMP Spin-Column	Tc-LAMP PrintrLab
OraL Chagas disease	oCD1	M	16	At the time of confirming the case (2009), IgM and IgG serology postive and no evidence of cardiac abnormality due to acute phase. Treatment with benznidazole 7 mg/kg/day for 60 days	GEB	oCD1.1	0	5.13	Positive	Positive	NP
						oCD1.2	12 m	1.97	Positive	Positive	NP
						oCD1.3	24 m	NAR	Negative	Negative	NP
	oCD2	M	8	At the time of confirming the case (2009), IgM and IgG serology postive. abnormal EKG. Treatment with benznidazole 7 mg/kg/day for 60 days.	GEB	oCD2.1	0	14.87	Positive	Positive	Positive
						oCD2.2	36 m	1.75	Positive	Positive	NP
						oCD2.3	108 m	1.6	Positive	Positive	NP
	oCD3	F	9	At the time of confirming the case (2009), IgM positive and IgG negative. abnormal EKG. Treatment with benznidazole 7 mg/kg/day for 60 days	GEB	oCD3.1	0	4586.77	Positive	Positive	NP
						oCD3.2	6 m	3521.09	Positive	Positive	NP
						oCD3.3	24 m	2258.83	Positive	Positive	NP
	oCD4	M	36	At the time of confirming the case (2009), IgM and IgG serology postive. abnormal EKG. Treatment with benznidazole 7 mg/kg/day for 60 days.	GEB	oCD4.1	0	1733.56	Positive	Positive	NP
						oCD4.2	36 m	1842.84	Positive	Positive	NP
						oCD4.3	108 m	1172.27	Positive	Positive	NP
	OCD5	F	7	At the time of confirming the case (2009), IgM and IgG serology postive. abnormal EKG. Treatment with benznidazole 7 mg/kg/day for 60 days.	GEB	oCD5.1	0	0.35	Positive	Negative	NP
						oCD5.2	36 m	NAR	Negative	Negative	NP
						oCD5.3	60 m	5	Positive	Positive	NP
	OCD6	F	9	At the time of confirming the case (2009), IgM aad IgG serology postive. abnormal EKG. Strout positive Treatment with benznidazole 7 mg/kg/day for 60 days.	GEB	oCD6.1	0	0.35	Positive	Positive	NP
						oCD6.2	36 m	5.56	Positive	Positive	Positive
						oCD6.3	60 m	14.99	Positive	Negative	Positive
	oCD7	F	10	At the time of confirming the case (2009), IgM and IgG serology postive. abnormal EKG. Treatment with benznidazole 7 mg/kg/day for 60 days	GEB	oCD7.1	12 m	1.75	Positive	Positive	Positive
						oCD7.2	108 m	2.93	Positive	Positive	NP
	oCD8	F	11	At the time of confirming the case (2009), IgM and IgG serology postive. abnormal EKG. Treatment with benznidazole 7 mg/kg/day for 60 days	GEB	oCD8.1	60 m	2697.31	Positive	Positive	NP
						oCD8.2	108 m	1126.81	Positive	Positive	NP
	oCD9	F	8	At the time of confirming the case (2009), IgM and IgG serology postive aNP no evidence of cardiac abnormality due to acute phase. Treatment with benznidazole 7 mg/kg/day/day for 60 days	GEB	oCD9.1	24 m	2404.02	Positive	Positive	NP
						oCD9.2	60 m	NAR	Negative	Negative	NP
AIDS-Chagas Reactivation	HIV-RCD1	M	42	Seizures/encephalitis with two space-occupying lesions CD4 7 cells/mL, Trypomastigotes in CSF	EB	HIV-RCD1.1	0	107	Positive	Positive	Positive
						HIV-RCD1.2	5 d	2	Positive	Positive	NP
						HIV-RCD1.3	14 d	NAR	Negative	Negative	NP
	HIV-RCD2	M	55	Sensory impairment/marked cerebral cortex atrophy; encephalitis with two space- occupying lesion CD4 10 cells/mL. Trypomastigotes in CSF	CSF	HIV-RCD2.1	0	3511.5	Positive	Positive	NP
					CSF	HIV-RCD2.2	7 d	13556	Positive	Positive	NP
					EB	HIV-RCD2.3	14 d	NAR	Negative	Negative	Negative
	HIV-RCD3	F	39	Right Hemiparesis, faciobrachiocrural/encephalytis with large space occupying lesion aNP brain midline shift CD4 10 cells/mL Strout Positive	EB	HIV-RCD3.1	0	677	Positive	Positive	NP
						HIV-RCD3.2	24 d	12,7	Positive	Positive	NP
						HIV-RCD3.3	38 d	NAR	Negative	Positive	NP

Grey boxes indicate the samples collected since initiation of treatment. No samples at initiation of treatment were available for oCD7-9. Tx: transplant, F: female; M: male; d, days; m, months; GEB: guanidine-EDTA Blood; EB: EDTA blood; CSF: cerebrospinal fluid; par.eq./mL: parasite equivalents per millilitre of sample; NA: not applicable; NAR: no amplification reaction, NP: not performed. ^#^ Parasite loads were measured in the original samples during the patients’ follow-up. ^§^: Qualitative qPCR results correspond to the archival paired sample retested by qPCR at the time Tc-LAMP was carried out in archival samples.

**Table 2 microorganisms-10-00909-t002:** Follow-up of Tx-RCD and Tc-RID clinical groups by qPCR and Tc-LAMP.

Clinical Group	ID Patient	Gender (F/M)	Age (Years)	Tx organ/Strout	Sample ID	Days before or after Transplant	Days from initial CD Treatment	^#^ Mean Parasite Load (par.eq/mL)	^§^ qPCR Result	Tc-LAMP Spin-Column	Tc-LAMP PrintrLab
Chagas Disease Reactivation	Tx-RCD1	M	71	Kidney/No Strout available.	Tx-RCD1.1	6	NA	1.86	Positive	NP	NP
					Tx-RCD1.2	14	NA	5.14	Positive	NP	NP
					Tx-RCD1.3	25	NA	13.65	Positive	Positive	Positive
					Tx-RCD1.4	28	0	59.14	Positive	Positive	Positive
					Tx-RCD1.5	35	7	NAR	Negative	Negative	Positive
	Tx-RCD2	M	61	Heart/No Strout available	Tx-RCD2.1	-54	NA	NAR	Negative	NP	NP
					Tx-RCD2.2	15	NA	0.17	Positive	NP	NP
					Tx-RCD2.3	286	NA	2.27	Positive	Positive	Positive
					Tx-RCD2.4	295	0	20.99	Positive	Positive	Positive
					Tx-RCD2.5	356	61	NAR	Negative	Negative	Negative
	Tx-RCD3	M	57	Liver/Positive Strout result 78 days after transplant	Tx-RCD3.1	7	NA	NAR	Negative	NP	NP
					Tx-RCD3.2	29	NA	12.11	Positive	NP	NP
					Tx-RCD3.3	33	NA	1.09	Positive	NP	NP
					Tx-RCD3.4	71	-7	0.81	Positive	Positive	Positive
					Tx-RCD3.5	92	14	NAR	Negative	Positive	Negative
	Tx-RCD4	M	57	Kidney/Positive Strout result 27 days after transplant	Tx-RCD4.1	6	NA	1.11	Positive	Positive	Positive
					Tx-RCD4.2	14	NA	82.16	Positive	Positive	Positive
					Tx-RCD4.3	34	7	269.97	Positive	Positive	Positive
					Tx-RCD4.4	55	28	NAR	Negative	Negative	Negative
	Tx-RCD5	F	66	Bone Marrow/No Strout available	Tx-RCD5.1	-2	0	16.78	Positive	Positive	Positive
					Tx-RCD5.2	6	8	NAR	Negative	Negative	Negative
					Tx-RCD5.3	22	24	NAR	Negative	Negative	Negative
Seropositive Donor Seronegative recipient transplant	Tx-RID	F	63	Liver and Kidney/No Strout available	Tx-RID.1	40	NA	0.49	Positive	Negative	NP
					Tx-RID.2	61	0	39.7	Positive	Positive	Positive
					Tx-RID.3	69	8	NAR	Negative	Negative	NP
					Tx-RID.4	141	80	NAR	Negative	Negative	Negative
					Tx-RID.5	155	94	NAR	Negative	Negative	NP
					Tx-RID.6	244	183	NAR	Negative	Negative	NP

Grey boxes indicate the samples collected since initiation of treatment. EB, EDTA-treated blood; d, days; m, months; NA, not applicable, NAR; No amplification reaction; NP: not performed; F: female, M: male; p.eq/mL, parasite equivalents per ml of sample. ^#^ Parasite loads were measured in the original samples during the patients’ follow-up. ^§^: Qualitative qPCR results correspond to the archival paired sample retested by qPCR at the time Tc-LAMP was carried out in archival samples.

## Data Availability

The data presented in this study are available at the URL https://cloud.dna.uba.ar/index.php/s/snOPIMZ0PO1z4by. The password is available under request from the corresponding author.

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
