# Peer review of "Loop-Mediated Isothermal Amplification of Trypanosoma cruzi DNA for Point-of-Care Follow-Up of Anti-Parasitic Treatment of Chagas Disease"

_microorganisms, 2022, doi:10.3390/microorganisms10050909_

Round 1

Reviewer 1 Report

The article presents results from the evaluation of LAMP as a surrogate marker of treatment failure in Chagas disease. The authors tested samples from the patients with acute or reactivated CD. Furthermore, as a comparator test the authors used qPCR published before.

In my opinion the communication is written in a very unclear way. At the first sight, it is difficult to understand how many samples were examined. Furthermore, novelty of the study is not so high. Additionally, the authors concentrate mainly for comparison LAMP with qPCR and implementation new extraction method PrintrLab extraction which became purpose of the study. I would suggest the authors change the tittle of the communication which is connected with changing discussion.

Thus the authors should consider or improved the manuscript based on the following suggestions:

Material and methods:

1.please assign a number to “Clinical samples” e.g 2.1, and similarly others: “DNA extraction” etc

2.the sentence “T. cruzi Loopamp kit (Tc-LAMP) was compared with standardized

qPCR using panels of archival clinical specimens collected from the following groups of

patients” should be replaced to the line 153.

3.please clarify why the kit was used for only some of the samples MagMaxMulti Sample Ultra 12.0 DNA extraction kit (Thermo Fisher, USA)?

Results:

1.In my opinion the authors should concentrate only on results from LAMP in the 159-168. There is no need here described symptoms of disease and results of EKG, because it is not aim of the study.

2.Please clarify how was the Cohen’s Kappa coefficient counted? And between which group? Whether the authors took into account only 6 samples when they compared Spin-column and PrintrLab DNA extractions? Or 33 vs 6?

3.line 246: please change “reference” to comparator

Author Response

Dear Sir,

Thank you for your comments.  Please find below our answers to your queries.

As suggested, the title of the communication has been changed to: 

Loop-Mediated Isothermal Amplification of Trypanosoma cruzi DNA for point-of -care follow-up of anti-parasitic treatment of Chagas Disease

Material and methods:

1.please assign a number to “Clinical samples” e.g 2.1, and similarly others: “DNA extraction” etc

Response: This has been done in the revised Tables.

2. the sentence “T. cruzi Loopamp kit (Tc-LAMP) was compared with standardized qPCR using panels of archival clinical specimens collected from the following groups of patients” should be replaced to the line 153.

Response: This sentence was replaced to the line 153 of the original version of the manuscript with a few changes. 

3. please clarify why the kit was used for only some of the samples MagMaxMulti Sample Ultra 12.0 DNA extraction kit (Thermo Fisher, USA)?

Response: The MagMax kit was used to perform DNA extractions in the PrintrLab device. We could only process the stored samples that had enough volume of blood for DNA extraction (n=23).

Results:

1.In my opinion the authors should concentrate only on results from LAMP in the 159-168. There is no need here described symptoms of disease and results of EKG, because it is not aim of the study.

Response: According to this comment, we have only left the information regarding EKG and disease symptoms in the Table.

2.Please clarify how was the Cohen’s Kappa coefficient counted? And between which group? Whether the authors took into account only 6 samples when they compared Spin-column and PrintrLab DNA extractions? Or 33 vs 6?

Response: The numbers of samples that were compared using the different methods ( LAMP-PrintrLab vs LAMP-spin column, LAMP-Printrlab vs qPCR and LAMP-spin-column vs qPCR ) are detailed in the revised version of the manuscript.

3.line 246: please change “reference” to comparator

Response: Done.

Reviewer 2 Report

Abstract: authors could add some description of time since treatment.

Line 31. Amend to '...0.71 to 1.00], respectively.'

Line 44. Change 'to HIV' to 'with HIV'.

Line 94. State the number of patients in this group (i.e.,. 5 patients).

Line 111. Authors should check if 19+4 = 24 as stated.

Line 111. Authors should explain the abbreviation GEB at its first mention in the text.

Figure1 legend. Should contain more information, for examples as given in lines 115-118.

Figure 2 (i). This shows novel data derived from this work, and so should be in the Results not Methods.

Figure 2 (ii). Part A is not clearly showing the LAMP positives, the authors should include a better photo or not include one.

Line 130. The authors should clarify if were the LAMP samples were judged positive by both naked eye AND UV observation, or just by one of these methods.

Line 176. Amend 'SpinSpin'

Table 1. Patient HIV-RCD3 has information not in bold font.

Line 191. Authors should add a number after the second use of Tx-RCD.

Line 211. Authors should amend RCD V to RCD 5, if this is correct.

Line 211. Authors should explain the abbreviation CDR at its first mention in the text.

Author Response

Dear Reviewer.

Thank you for your comments. Please, find the responses to them below:

Abstract: authors could add some description of time since treatment.

Response: We could not describe the time since treatment in the abstract, because this was different for each clinical group, and even among patients from a same clinical group. Thus, the times since treatment for each case are described in the Tables. 

Line 31. Amend to '...0.71 to 1.00], respectively.'

Response: Done

Line 44. Change 'to HIV' to 'with HIV'.

Response: Done

Line 94. State the number of patients in this group (i.e.,. 5 patients).

Response: Done

Line 111. Authors should check if 19+4 = 24 as stated.

Response: Apologizes for this. 23 is the right number

Line 111. Authors should explain the abbreviation GEB at its first mention in the text.

Response. Done

Figure1 legend. Should contain more information, for examples as given in lines 115-118.

Response: Done 

Figure 2 (i). This shows novel data derived from this work, and so should be in the Results not Methods.

Response: The Figure 2 shows an example of the results obtained in this work; it was replace in the Results section.

Figure 2 (ii). Part A is not clearly showing the LAMP positives, the authors should include a better photo or not include one.

Response: The photo was replaced for a better one, including the visualization of the LAMP microtubes before amplification, so the difference between positives and negatives can be better appreciated. 

Line 130. The authors should clarify if were the LAMP samples were judged positive by both naked eye AND UV observation, or just by one of these methods.

Response: The samples were judged positive by both observations, with complete agreement. This was added in the new version of the manuscript.

Line 176. Amend 'SpinSpin'

Response. Done

Table 1. Patient HIV-RCD3 has information not in bold font.

Response. Corrected

Line 191. Authors should add a number after the second use of Tx-RCD.

Response. Done

Line 211. Authors should amend RCD V to RCD 5, if this is correct.

Response. Done

Line 211. Authors should explain the abbreviation CDR at its first mention in the text.

Response: Chagas disease reactivation, done.

Round 2

Reviewer 1 Report

Dear Authors,

After applying corrections I think that the article in its current form can be published in Microorganisms.

I have nothing to add.